# MVSMamba: Multi-View Stereo with State Space Model

**Jianfei Jiang**    **Qiankun Liu***   **Hongyuan Liu**    **Haochen Yu**
**Liyong Wang**    **Jiansheng Chen**    **Huimin Ma***
University of Science and Technology Beijing, China
{jiangjf,hongyuanliu,haochen.yu,wangly}@xs.ustb.edu.cn
{liuqk3,jschen,mhmpub}@ustb.edu.cn

## Abstract

Robust feature representations are essential for learning-based Multi-View Stereo (MVS), which relies on accurate feature matching. Recent MVS methods leverage Transformers to capture long-range dependencies based on local features extracted by conventional feature pyramid networks. However, the quadratic complexity of Transformer-based MVS methods poses challenges to balance performance and efficiency. Motivated by the global modeling capability and linear complexity of the Mamba architecture, we propose MVSMamba, the first Mamba-based MVS network. MVSMamba enables efficient global feature aggregation with minimal computational overhead. To fully exploit Mamba's potential in MVS, we propose a Dynamic Mamba module (DM-module) based on a novel reference-centered dynamic scanning strategy, which enables: (1) Efficient intra- and inter-view feature interaction from the reference to source views, (2) Omnidirectional multi-view feature representations, and (3) Multi-scale global feature aggregation. Extensive experimental results demonstrate MVSMamba outperforms state-of-the-art MVS methods on the DTU dataset and the Tanks-and-Temples benchmark with both superior performance and efficiency. The source code is available at https://github.com/JianfeiJ/MVSMamba.

## 1   Introduction

Multi-View Stereo (MVS) is aims to reconstruct dense 3D geometry of objects or scenes from calibrated multi-view images, which is widely used in fields like autonomous driving [1, 2]. It estimates the depth of each pixel by identifying correspondences across source views that satisfy multi-view geometric consistency, making the task highly dependent on the quality of feature matching. Naturally, more robust feature representations lead to more reliable feature matching.

Traditional MVS methods [3, 4, 5, 6, 7, 8] rely on handcrafted features, which tend to perform poorly in regions with repetitive patterns, weak textures, and reflections. In contrast, learning-based MVS methods [9, 10, 11] have made significant strides by leveraging the strong representational power of deep neural networks. Early learning-based MVS methods utilize Convolutional Neural Networks (CNNs)[12] and their variants[13, 14, 15, 16] for feature extraction, but their limited receptive fields restricted to local features only. Recently, some methods have introduced Transformers [17] to model long-range dependencies, enabling global feature learning both within and across views, thus improving the robustness of feature matching in challenging regions.

Despite significant progress, Transformer-based MVS methods [18, 19, 20] continue to suffer from quadratic computational complexity. To mitigate this issue, existing approaches have introduce linear

---

*Corresponding authors

39th Conference on Neural Information Processing Systems (NeurIPS 2025).

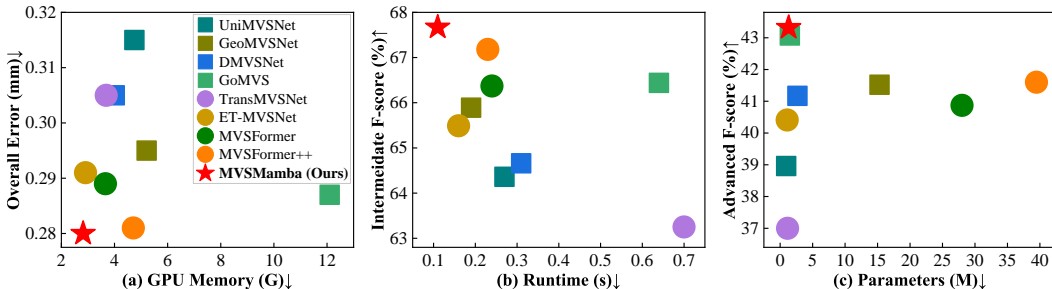

Figure 1: Comparison of **performance vs. efficiency** among state-of-the-art CNN-based (■), Transformer-based (●), and our Mamba-based (★) methods on the (a) DTU dataset, the Tanks-and-Temples (b) intermediate and (c) advanced benchmark. The GPU memory and runtime are evaluated on 5-view images with a resolution of 832×1152. The proposed MVSMamba achieves the best performance with superior efficiency.

attention [18, 21], epipolar window attention [22, 23], and epipolar vanilla attention [17, 24], and typically applying them at the lowest resolution stage to reduce computational cost. However, these methods still involve multiple rounds of self-attention and cross-attention across both reference and source features, resulting in substantial overhead. Consequently, striking an optimal balance between performance and efficiency remains a major challenge in MVS. A critical question remains: *How can we sustain high performance while minimizing computational cost?*

As a powerful variant of state space models [25], Mamba [26] offers a promising solution by enabling effective modeling of long-range dependencies with linear complexity. Inspired by this, we propose MVSMamba, the **first** MVS network to explore the Mamba architecture. MVSMamba is designed to efficiently model long-range dependencies among multi-view features, addressing performance and efficiency bottlenecks in challenging MVS scenarios. To incorporate Mamba into the multi-view feature matching process, we introduce a novel Dynamic Mamba module (DM-module) based on a *reference-centered dynamic scanning* strategy. This module facilitates the efficient learning of global and omnidirectional feature representations across multiple views. Specifically, for each reference-source feature pair, the source features are concatenated to the top, bottom, left, and right of the reference features, enabling **four directional** scanning from the reference view toward the source view. This spatial configuration enhances intra- and inter-view interactions for each feature pair. To generalize beyond pairwise matching, we **dynamically** adjust the scan directions based on the index of source views, performing **omnidirectional** scanning from the reference feature toward different source views. This enables the construction of omnidirectional multi-view feature representations. The DM-module is deployed at multiple scales within the FPN to capture long-range dependencies across different spatial resolutions. As shown in Fig. 1, CNN-based methods [11, 27, 28, 29] are relatively efficient but suffer from limited performance. Transformer-based methods [18, 24, 30, 20] offer superior performance, but this often comes at the cost of reduced efficiency. In contrast, the proposed MVSMamba achieves state-of-the-art performance on both the DTU dataset and the Tanks-and-Temples benchmark, while offering superior efficiency.

The main contributions are summarized as follows:

- We present MVSMamba, the first MVS network to leverage the Mamba architecture, enabling efficient global and omnidirectional multi-view feature representation.
- We propose a novel Dynamic Mamba module based on a reference-centered dynamic scanning strategy to effectively bridge directional scanning with multi-view feature aggregation.
- Extensive experiments on the DTU dataset and the Tanks-and-Temples benchmark demonstrate that MVSMamba achieves state-of-the-art performance with superior efficiency.

## 2 Related Work

### 2.1 Learning-based MVS

In recent years, learning-based MVS mehthods [9, 31] have made significant progress compare with traditional methods. MVSNet [9] presents the first end-to-end learning-based MVS method,

leveraging Convolutional Neural Network (CNN) for feature extraction. Subsequent methods have introduced improvements from various perspectives. RNN-based MVS methods [10, 32] reduces memory consumption but suffers from slow inference speed. Iterative update-based MVS methods [33, 34, 35, 36, 37] enabling high efficiency but with limited performance. Coarse-to-fine MVS methods [11, 38, 39] achieves relatively better trade-off between performance and efficiency, and have gradually become the dominant paradigm.

Coarse-to-fine MVS methods typically employ Feature Pyramid Networks [12] (FPN) to extract multi-scale features, enabling depth estimation at multiple resolutions. Some later works [14, 18] adopt CNN variants [15] to enhance feature representations. However, due to the limited receptive field of CNNs, these methods are constrained to capture local features. To address this, TransMVS-Net [18] first introduces Transformers [17] to MVS, employing intra- and inter- view attention [21] to aggregate global features. WT-MVSNet [22] incorporates Swin Transformer [23] and constraines feature aggregation within epipolar-aligned windows. ET-MVSNet [24] further restricts vanilla attention [17] to epipolar line pairs, enabling non-local feature aggregation. Moreover, MVSFormer [30], MVSFormer++ [20] and MonoMVSNet [40] enhance FPN features using pretrained Vision Transformers [41, 42, 43] (ViT). Although these Transformer-based MVS methods have made efforts to address the complexity issues inherent in attention mechanisms, they either inevitably require alternating computations of self-attention and cross-attention to construct long-range dependency, or rely on parameter-heavy pretrained models, making it difficult to simultaneously achieve high performance and efficiency.

## 2.2 State Space Models

Transformers [17] have substantially advanced in computer vision but are hindered by their quadratic complexity. To mitigate this limitation, researchers have developed more efficient alternatives, including linear attention [21], shifted window attention [23], and flash attention [44]. Concurrently, state space models [25], combined with selective mechanisms, have gained traction for capturing long-range dependencies with linear complexity (detailed in Appendix A). Recently, Mamba [26] has shown promising performance in computer vision tasks [45, 46, 47]. Vim [48] and VMamba [49] expand receptive fields globally using bidirectional and four-directional scanning, respectively. EVMamba [50] introduces a skip-scan mechanism to improve computational efficiency. Subsequent works, including MambaVision [51], EfficientViM [52], and Mamba-ND[53], further explored this domain by combining Mamba with self-attention, reducing computational costs, or extending the architecture to multi-dimensional data. JamMa [54] proposes Joint Mamba for feature matching, which enables high-frequency interactions between feature pairs. Building on these advances, we integrate Mamba into the one-to-many multi-view stereo (MVS) setting to capture long-range dependencies across multi-view features. This novel adaptation is specifically designed to address the unique challenges of MVS.

## 3 Methodology

### 3.1 Network Overview

The overall architecture of MVSMamba is depicted in Fig. 2. Given $K$ input images $\{\mathbf{I}_k\}_{k=0}^{K-1} \in \mathbb{R}^{3 \times H \times W}$ consist of a reference image ($k = 0$) and $K - 1$ source images ($0 < k < K$), the goal is to estimate a depth map for the reference image. We integrate the proposed Dynamic Mamba module (DM-module) into the conventional Feature Pyramid Network (FPN) [12] to capture long-range dependencies across multi-view features, efficiently aggregating the local features of the FPN encoder into global and omnidirectional features. Then, we perform multi-scale aggregation within the FPN. Finally, we predicted depth map from the FPN output features in a coarse-to-fine manner [11, 55].

### 3.2 Dynamic Mamba Module

The feature aggregation paradigm in existing Transformer-based MVS methods [18, 24, 20] typically involves aggregating information from reference feature into source features [18], thereby enabling improved source feature representation. However, this process often requires repeatedly alternating between self-attention and cross-attention computations, making it difficult to achieve a favorable balance between performance and efficiency. Therefore, we propose a Dynamic Mamba module (DM-

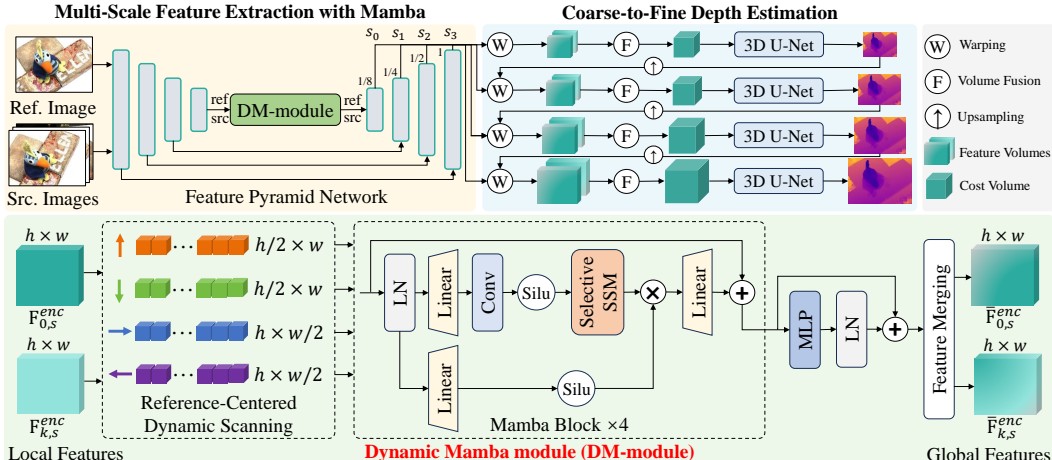

Figure 2: Overall architecture of MVSMamba. The proposed Dynamic Mamba module (DM-module) is integrated into the FPN (Sec. 3.2). First, a reference-centered dynamic scanning strategy extracts four directional feature sequences, which are processed by four independent Mamba blocks. The resulting sequences are then merged back into 2D feature maps. Multi-scale feature aggregation (Sec. 3.3) is subsequently performed. Finally, we predicted depth in a coarse-to-fine manner (Sec. 3.4).

module) with a novel reference-centered dynamic scanning strategy, which efficiently performs both intra- and inter-view omnidirectional global feature aggregation. Specifically, given FPN encoder features $\{\mathbf{F}_{k,s}^{enc} \in \mathbb{R}^{C \times \frac{H}{2^{3-s}} \times \frac{W}{2^{3-s}}} | s = 0, 1, 2, 3\}_{k=0}^{K-1}$, where $s$ is the scale index, DM-module leverages dynamic scanning order for feature enhancement.

**Reference-Centered Dynamic Scanning.** To construct long-range dependencies from reference feature to each source feature, we propose a reference-centered dynamic scanning strategy. As illustrated in Fig. 3 (a), take the $s$-th scale for example, source features $\mathbf{F}_{k,s}^{enc}$ are concatenated around the reference feature $\mathbf{F}_{0,s}^{enc}$ along both horizontal and vertical directions, placing them on the top, bottom, left, and right of the reference feature:

$$\mathbf{X}_{k,s}^{hr} = \left[\mathbf{F}_{0,s}^{enc} \mid \mathbf{F}_{k,s}^{enc}\right], \mathbf{X}_{k,s}^{hl} = \left[\mathbf{F}_{k,s}^{enc} \mid \mathbf{F}_{0,s}^{enc}\right], \mathbf{X}_{k,s}^{vb} = \left[\begin{array}{c}\mathbf{F}_{0,s}^{enc}\\\mathbf{F}_{k,s}^{enc},\end{array}\right], \mathbf{X}_{k,s}^{vt} = \left[\begin{array}{c}\mathbf{F}_{k,s}^{enc}\\\mathbf{F}_{0,s}^{enc}\end{array}\right], \quad (1)$$

where the concatenated features $\mathbf{X}_{k,s}^{hr}, \mathbf{X}_{k,s}^{hl} \in \mathbb{R}^{C \times \frac{H}{2^{3-s}} \times \frac{2W}{2^{3-s}}}$ are the horizontal-right, horizontal-left arrangement of features, and $\mathbf{X}_{k,s}^{vb}, \mathbf{X}_{k,s}^{vt} \in \mathbb{R}^{C \times \frac{2H}{2^{3-s}} \times \frac{W}{2^{3-s}}}$ are the vertical-top, and vertical-bottom arrangements, respectively.

Based on $\mathbf{X}_{k,s}^{hr}, \mathbf{X}_{k,s}^{hl}, \mathbf{X}_{k,s}^{vb}$, and $\mathbf{X}_{k,s}^{vt}$, we adopt the skip scanning [50] strategy with four different directions. As shown in Fig. 3 (a), these four concatenated features ensure the scanning order from the reference feature to the source feature, thereby capturing global contextual dependencies from the reference features to the source feature. In addition to global aggregation within both the reference and source features, the source feature can effectively learn global representation from the reference feature.

Specifically, we perform ⇗ order scanning on $\mathbf{X}_{k,s}^{hr}$, ⇖ order scanning on $\mathbf{X}_{k,s}^{hl}$, ⇙ order scanning on $\mathbf{X}_{k,s}^{vb}$ and ⇘ order scanning on $\mathbf{X}_{k,s}^{vt}$ with a step 2 and a dynamic starting coordinate $(h_k, w_k)$, resulting in four directional sequences $\{\mathbf{S}_{k,s}^j \in \mathbb{R}^{C \times \frac{HW}{2^{2(3-s)-1}}}\}_{j=1}^4$, each with a length of $\frac{HW}{2^{2(3-s)-1}}$:

$$\begin{aligned}\mathbf{S}_{k,s}^1 &= ⇗\left(\mathbf{X}_{k,s}^{hr}, (h_k, w_k)\right), \quad \mathbf{S}_{k,s}^2 = ⇖\left(\mathbf{X}_{k,s}^{hl}, (h_k, w_k)\right),\\\mathbf{S}_{k,s}^3 &= ⇙\left(\mathbf{X}_{k,s}^{vb}, (h_k, w_k)\right), \quad \mathbf{S}_{k,s}^4 = ⇘\left(\mathbf{X}_{k,s}^{vt}, (h_k, w_k)\right),\end{aligned} \quad (2)$$

The starting coordinate $(w_k, h_k)$ is dynamically updated according to the source image index $k$ and the arrangement type of reference and source features, as shonw in Fig. 3 (b). Specifically:

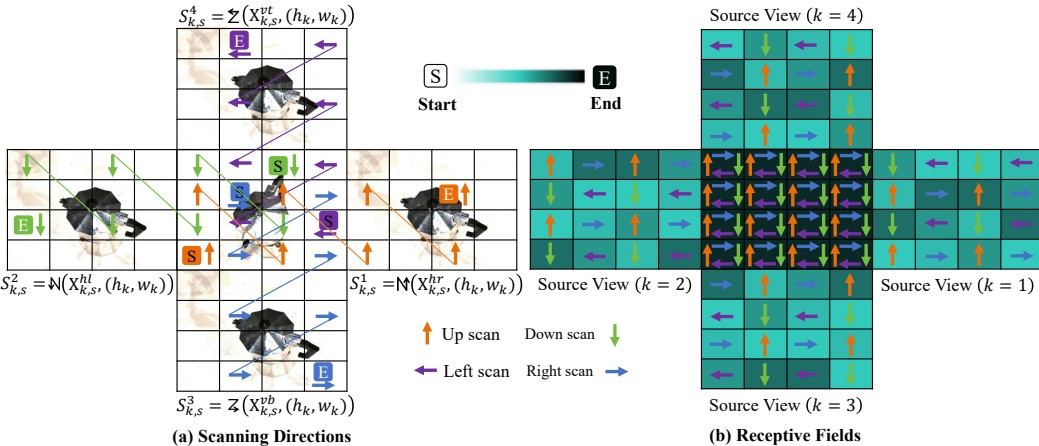

Figure 3: Overview of our proposed reference-centered dynamic scanning strategy. (a) Scanning directions of each reference-source feature pairs. (b) Receptive Filed of the reference feature to different source features.

$$h_k = (\lfloor \tfrac{j-1}{2} \rfloor + h_j) \bmod 2, \qquad\qquad \text{s.t.} \quad j = (k-1) \bmod 4, \text{and } (h_j, w_j) = \begin{cases} (1,0), & j = 0, \\ (0,0), & j = 1, \\ (0,1), & j = 2, \\ (1,1), & j = 3. \end{cases}$$
$$w_k = ((j-1) \bmod 2 + w_j) \bmod 2,$$
(3)

These directional sequences are then fed into four independent Mamba blocks [26] to establish long-range dependency:

$$\{\hat{\mathbf{S}}_j\}_{j=1}^4 = \mathrm{Mamba}(\{\mathbf{S}_j\}_{j=1}^4).$$
(4)

Finally, a Multilayer Perceptron (MLP) is introduced to further enhance the global representations of the four directional sequences:

$$\{\bar{\mathbf{S}}_j\}_{j=1}^4 = \{\hat{\mathbf{S}}_j\}_{j=1}^4 + \mathrm{LN}(\mathrm{MLP}(\{\hat{\mathbf{S}}_j\}_{j=1}^4)).$$
(5)

**Analysis of Dynamic Scanning.** With the dynamic scanning on different features, we can get omnidirectional global receptive field, enhancing the features effectively. Here we give a deeper analysis of our design.

Given a specific source image $k$, the source features are arranged around the reference feature. The scanning types with different orders and starting coordinates are shown in Fig. 3(a). As we can see, the scanning step 2 makes the length of scanned sequences 4 times smaller than the total number of pixels, which is beneficial to improve the efficiency of the proposed MVSMamba. However, for each scanning order, the features in most pixels are skipped, resulting in a smaller receptive field in the reference feature. Nonetheless, the different starting coordinates of different scanning orders ensure that all the pixels in the reference feature are scanned, resulting in a global receptive field in the reference feature.

Though a global receptive field in the reference feature is obtained when given a specific source image, the scanning in the reference feature is limited to a single direction for each pixel, resulting in a lack of omnidirectionality in the aggregated features. Since MVS is inherently a one-to-many feature matching task, where the reference feature typically needs to be matched with different source features, we can *change the scanning direction in the reference feature* for different source features, as illustrated in Fig. 3 (b), where 4 source features are available ($K \geq 5$). In our implementation, we choose to dynamically update the starting coordinates for different source features rather than changing the scanning direction directly, which produces an equivalent effect.

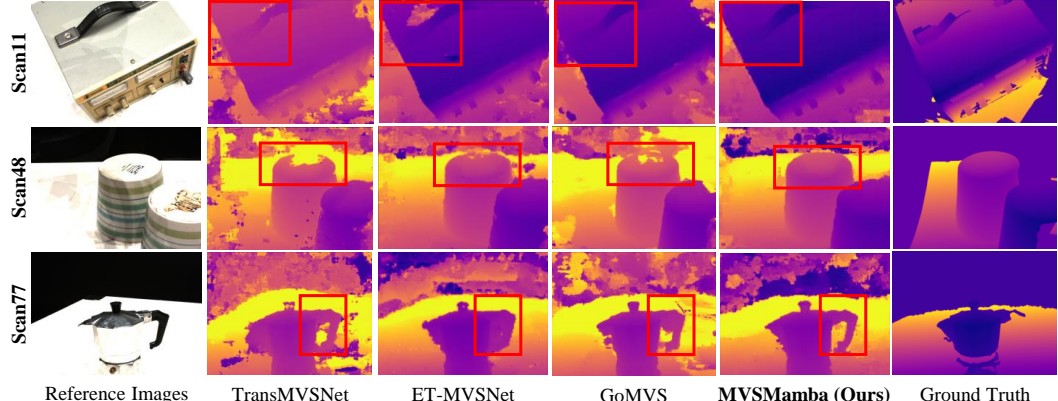

| Reference Images | TransMVSNet | ET-MVSNet | GoMVS | **MVSMamba (Ours)** | Ground Truth |

Figure 4: Qualitative comparison of depth maps in challenging scenarios on the DTU evaluation dataset. Our method predicted more accurate depth maps in texture-less and reflection regions.

**Feature Merging from Different Sequences.** After obtaining the enhanced feature sequences $\{\bar{\mathbf{S}}_j\}_{j=1}^4$ with long-range dependency, we need to recover the globally omnidirectional feature map for each reference–source feature pair by merging the enhanced features.

First, we rearrange the features in $\{\bar{\mathbf{S}}_j\}_{j=1}^4$ to four features $\bar{\mathbf{X}}_{k,s}^{hr}, \bar{\mathbf{X}}_{k,s}^{hl} \in \mathbb{R}^{C \times \frac{H}{2^{3-s}} \times \frac{2W}{2^{3-s}}}$, and $\bar{\mathbf{X}}_{k,s}^{vb}, \bar{\mathbf{X}}_{k,s}^{vt} \in \mathbb{R}^{C \times \frac{2H}{2^{3-s}} \times \frac{W}{2^{3-s}}}$ by inversing the scan operations. The enhanced reference feature $\bar{\mathbf{F}}_{0,s}^{enc}$ and source feature $\bar{\mathbf{F}}_{k,s}^{enc}$ is obtained as follows:

$$\bar{\mathbf{F}}_{0,s}^{enc} = \bar{\mathbf{X}}_{k,s}^{hr}[:,:,:\frac{W}{2^{3-s}}] + \bar{\mathbf{X}}_{k,s}^{hl}[:,:,\frac{W}{2^{3-s}}:] + \bar{\mathbf{X}}_{k,s}^{vb}[:,\frac{H}{2^{3-s}}:,:] + \bar{\mathbf{X}}_{k,s}^{vt}[:,:\frac{H}{2^{3-s}},:]$$

$$\bar{\mathbf{F}}_{k,s}^{enc} = \bar{\mathbf{X}}_{k,s}^{hr}[:,:,\frac{W}{2^{3-s}}:] + \bar{\mathbf{X}}_{k,s}^{hl}[:,:,:\frac{W}{2^{3-s}}] + \bar{\mathbf{X}}_{k,s}^{vb}[:,:\frac{H}{2^{3-s}},:] + \bar{\mathbf{X}}_{k,s}^{vt}[:,\frac{H}{2^{3-s}}:,:]$$

(6)

## 3.3 Multi-Scale Aggregation

The proposed DM-module takes both the reference-view and source-view features as input and enhances the features across the two views. Though it is effective, it brings some computational overhead. In order to reduce the computational complexity and achieve better efficiency, we further design a Simplified DM-module (SDM-module), and use these two modules at different scales.

The SDM-module only takes the reference or source feature as input and enhances the feature within the provided view. Since only a single-view feature is provided, we directly scan the input feature to produce four sequences similar to Eq. (2). The starting coordinates are obtained using Eq. (3) with $k = 1$. After feeding the sequences to Mamba blocks [26], the enhanced feature can be obtained by inversing the scan operations.

Given the DM-module and SDM-module, we only use DM-moudle at the 0-th scale. While decoding the enhanced $\{\bar{\mathbf{F}}_{k,0}^{enc}\}_{k=0}^{K-1}$ to multi-scale pyramid features, we insert a SDM-module before the output layer for the 1-st scale in the FPN decoder. The output features of the FPN decoder are denoted as $\{\bar{\mathbf{F}}_{k,s}^{dec} \in \mathbb{R}^{C \times \frac{H}{2^{3-s}} \times \frac{W}{2^{3-s}}} | s = 0, 1, 2, 3\}_{k=0}^{K-1}$.

## 3.4 Learning Depth from FPN Features

Based on the FPN output features, we predicted depth map in a coarse-to-fine manner [11]. First, source features are warped [9] into the reference view to form feature volumes [9], enabling the construction of pairwise reference–source feature similarities [56, 33]. These feature similarities are then fused into a cost volume using attention-based weights [55]. Subsequently, a lightweight 3D U-Net [55] is employed for cost volume regularization, followed by a softmax operation to generate a probability volume. Finally, a winner-take-all strategy is used to predict the depth map. For more details about coarse-to-fine depth estimation, please refer to the works in [11, 55]. Similar to existing MVS works [55], we apply cross-entropy loss at each scale to supervise the probability volume.

Table 1: Quantitative results of point cloud error and model efficiency on the DTU evaluation set with coarse-to-fine learning-based MVS methods. The methods are categorized into three groups (from top to bottom): CNN-based, Transformer-based, and our Mamba-based. Methods with * denotes trained on high-resolution images. To indicate the performance–efficiency balance, we report the average ranking across six metrics of point cloud error and model efficiency. The best , second-best , and third-best results are marked with colors.

| Methods | Avg. Rank↓ | Point Cloud Error↓ | | | Model Efficiency↓ | | |
| --- | --- | --- | --- | --- | --- | --- | --- |
| | | Overall | Acc. | Comp. | GPU(G) | Time(s) | Params(M) |
| CasMVSNet [11] | 7.17 | 0.355 | 0.324 | 0.385 | 4.48 | 0.18 | 0.93 |
| UniMVSNet [57] | 8.83 | 0.315 | 0.352 | 0.278 | 4.75 | 0.27 | 0.93 |
| MVSTER* [55] | 5.17 | 0.303 | 0.340 | 0.266 | 2.70 | 0.07 | 0.98 |
| GeoMVSNet [27] | 8.00 | 0.295 | 0.331 | 0.259 | 5.21 | 0.19 | 15.31 |
| DMVSNet [28] | 8.83 | 0.305 | 0.338 | 0.272 | 4.01 | 0.31 | 2.67 |
| GoMVS [29] | 7.67 | 0.287 | 0.347 | 0.227 | 12.1 | 0.64 | 1.50 |
| TransMVSNet [18] | 7.83 | 0.305 | 0.321 | 0.289 | 3.69 | 0.70 | 1.15 |
| ET-MVSNet [24] | 5.00 | 0.291 | 0.329 | 0.253 | 2.91 | 0.16 | 1.09 |
| MVSFormer* [30] | 6.17 | 0.289 | 0.327 | 0.251 | 3.66 | 0.24 | 28.01 |
| MVSFormer++* [20] | 5.83 | 0.281 | 0.309 | 0.252 | 4.71 | 0.23 | 39.48 |
| **MVSMamba (Ours)** | 3.83 | 0.287 | 0.314 | 0.260 | 2.82 | 0.11 | 1.31 |
| **MVSMamba* (Ours*)** | 2.50 | 0.280 | 0.308 | 0.252 | 2.82 | 0.11 | 1.31 |

# 4 Experiment

## 4.1 Datasets

We conduct experiments on three of the most widely used datasets in the field of MVS. (1) **DTU** [58] is an indoor dataset captured under controlled laboratory conditions, consisting of 128 scenes. Each scene is captured under seven different lighting conditions with either 49 or 64 images. Following the MVSNet [9] protocol, we split the dataset into training, validation, and evaluation sets, resulting in a total of 27,097 training samples. DTU used for both training and evaluation. (2) **Tanks-and-Temples** [59] is a large-scale benchmark captured in real-world environments, containing 14 indoor and outdoor scenes. The dataset is divided into a intermediate set and a advanced set based on reconstruction difficulty, and is used to evaluation the generalization ability of MVS methods. (3) **BlendedMVS** is a large-scale synthetic dataset MVS training only, comprising 106 training scenes and 7 validation scenes.

## 4.2 Implementation Details

MVSMamba is implemented using PyTorch [60] and optimized with the Adam optimizer [61]. Following common practice [55, 30, 20], the model is first trained and evaluated on the DTU [58] dataset. To adapt the model to real-world scenes, the DTU-trained model is fine-tuned on the BlendedMVS [62] dataset before evaluation on the Tanks-and-Temples benchmark [59]. The final reconstructed point clouds are obtained using the dynamic fusion strategy [32].

For DTU training, we use 5-view input images at a resolution of $512 \times 640$, with a batch size of 4 for 15 epochs. The initial learning rate is set to 0.001 and is halved at the 10-th, 12-th, and 14-th epochs. For fine-tuning on BlendedMVS, we use 11-view images at a resolution of $576 \times 768$ with a batch size of 2 for 15 epochs. The initial learning rate is 0.0005 and is reduced by half at the 6-th, 8-th, 10-th, and 12-th epochs. Additionally, consistent with [55, 30, 20], we conduct high-resolution training on DTU using 5-view images at $1024 \times 1280$ resolution for 10 epochs, with an initial learning rate of 0.001, halved at 6-th, 8-th, and 9-th epochs. The number of inverse depth hypotheses in four coarse-to-fine scales is set to 32-16-8-4, with corresponding depth intervals of 2-1-1-0.5, and the group correlation of 4-4-4-4.

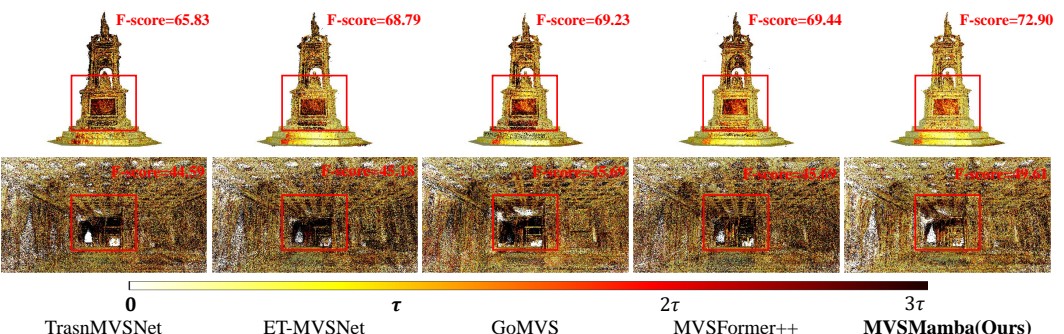

Figure 5: Qualitative comparison of reconstructed point clouds on the Tanks-and-Temples benchmark. The top row shows the precision of Francis ($\tau = 5mm$) from the intermediate set, while the bottom row presents the precision of Ballroom ($\tau = 10mm$) from the advanced set. Brighter regions indicate lower reconstruction errors under the corresponding distance threshold $\tau$.

Table 2: Quantitative results on the Tanks-and-Temples benchmark with F-score [%]. The **mean** refers the average F-score of all scenes. Methods are categorized into three groups (from top to bottom): CNN-based, Transformer-based, and our Mamba-based. The best , second-best , and third-best results are marked with colors.

| Methods | Intermediate set ↑ | | | | | | | | | Advanced set ↑ | | | | | | |
|---|---|---|---|---|---|---|---|---|---|---|---|---|---|---|---|---|
| | **Mean** | Fam. | Fra. | Hor. | L.H. | M60 | Pan. | P.G. | Tra. | **Mean** | Aud. | Bal. | Cou. | Mus. | Pal. | Tem. |
| CasMVSNet [11] | 56.84 | 76.37 | 58.45 | 46.26 | 55.81 | 56.11 | 54.06 | 58.18 | 49.51 | 31.12 | 19.81 | 38.46 | 29.10 | 43.87 | 27.36 | 28.11 |
| UniMVSNet [57] | 64.36 | 81.20 | 66.43 | 53.11 | 63.46 | 66.09 | 64.84 | 62.23 | 57.53 | 38.96 | 28.33 | 44.36 | 39.74 | 52.89 | 33.80 | 34.63 |
| MVSTER [55] | 60.92 | 80.21 | 63.51 | 61.38 | 61.47 | 58.16 | 58.98 | 51.38 | | 37.53 | 26.68 | 42.14 | 35.65 | 49.37 | 32.16 | 39.19 |
| GeoMVSNet [27] | 65.89 | 81.64 | 67.53 | 55.78 | 68.02 | 65.49 | 67.19 | 63.27 | 58.22 | 41.52 | 30.23 | 46.54 | 39.98 | 53.05 | 35.98 | 43.34 |
| DMVSNet [28] | 64.66 | 81.27 | 67.54 | 59.10 | 63.12 | 64.64 | 64.80 | 59.83 | 56.97 | 41.17 | 30.08 | 46.10 | 40.65 | 53.53 | 35.08 | 41.60 |
| GoMVS [29] | 66.44 | 82.68 | 69.23 | 69.19 | 63.56 | 65.13 | 62.10 | 58.81 | 60.80 | 43.07 | 35.52 | 47.15 | 42.52 | 52.08 | 36.34 | 44.82 |
| TransMVSNet [18] | 63.52 | 80.92 | 65.83 | 56.94 | 62.54 | 63.06 | 60.00 | 60.20 | 58.67 | 37.00 | 24.84 | 44.59 | 34.77 | 46.49 | 34.69 | 36.62 |
| CostFormer [63] | 64.51 | 81.31 | 65.65 | 55.57 | 63.46 | 66.24 | 65.39 | 61.27 | 57.30 | 39.43 | 29.18 | 45.21 | 39.88 | 53.38 | 34.07 | 34.87 |
| WT-MVSNet [22] | 65.34 | 81.87 | 67.33 | 57.76 | 64.77 | 65.68 | 64.61 | 62.35 | 58.38 | 39.91 | 29.20 | 44.48 | 39.55 | 53.49 | 34.57 | 38.15 |
| ET-MVSNet [24] | 65.49 | 81.65 | 68.79 | 59.46 | 65.72 | 64.22 | 64.03 | 61.23 | 58.79 | 40.41 | 28.86 | 45.18 | 38.66 | 51.10 | 35.39 | 43.23 |
| MVSFormer [30] | 66.37 | 82.06 | 69.34 | 60.49 | 68.61 | 65.67 | 64.08 | 61.23 | 59.53 | 40.87 | 28.22 | 46.75 | 39.30 | 52.88 | 35.16 | 42.95 |
| MVSFormer++ [20] | 67.18 | 82.69 | 69.44 | 64.24 | 69.16 | 64.13 | 66.43 | 61.19 | 60.12 | 41.60 | 29.93 | 45.69 | 39.46 | 53.58 | 35.56 | 45.39 |
| **MVSMamba (Ours)** | 67.67 | 82.47 | 72.90 | 58.55 | 69.63 | 65.34 | 66.88 | 65.60 | 59.98 | 43.32 | 30.95 | 49.61 | 41.04 | 54.92 | 36.72 | 46.67 |

## 4.3 Benchmark Performance

**Evaluation on DTU.** We use the official evaluation script to report three standard metrics: accuracy (Acc.), completeness (Comp.), and their average (Overall). Moreover, we also evaluate the model efficiency (GPU memory, runtime and parameters) using 5-view input images with resolution of $832 \times 1152$ to ensure fair comparison. For the model trained on low-resolution (MVSMamba), we use 5-view input images at a resolution of $832 \times 1152$. For the model trained on high-resolution (MVSMamba*), we use 5-view input images with a resolution of $1152 \times 1600$. The quantitative results of point cloud error and model efficiency are shown in Tab. 1. We compare our method with state-of-the-art coarse-to-fine learning-based MVS methods. MVSMamba* achieves the highest overall score and accuracy, while also demonstrating the best balance between performance and efficiency. Meanwhile, MVSMamba outperforms all other methods in the performance-efficiency trade-off, with performance second only to MVSFormer++[30], which was trained on high-resolution images with the lower efficiency. As shown in Fig. 4, our method produces more accurate depth maps in challenging regions, highlighting its robustness and generalization capability.

**Evaluation on Tanks-and-Temples.** We evaluate our method on the Tanks-and-Temples benchmark to assess its generalization capability, and report the F-score as the metric. Consistency with [30, 20], the evaluation is conducted using 21-view input images with 2k resolution. The quantitative results of intermediate and advanced sets are shown in Tab. 2. Our method achieves best performance on both intermediate and advanced sets among all published methods, which demonstrate our powerful generalization capability. Fig. 5 shows the qualitative results of reconstructed point clouds, our method exhibit superior precision. More visualization results are provided in Appendix D.

Table 3: Ablation study of each component in MVSMamba.

| Modules | Overall↓ | Acc.↓ | Comp.↓ | MAE↓ | GPU(G)↓ | Time(s)↓ | Params(M)↓ |
|---------|----------|-------|--------|------|---------|----------|------------|
| full | 0.287 | 0.314 | 0.260 | 5.21 | 2.82 | 0.111 | 1.31 |
| w/o DM | 0.295 | 0.317 | 0.272 | 5.58 | 2.82 | 0.104 | 1.15 |
| w/o SDM | 0.289 | 0.317 | 0.261 | 5.45 | 2.82 | 0.097 | 1.15 |
| w/o MLP | 0.293 | 0.315 | 0.271 | 5.23 | 2.82 | 0.108 | 1.24 |

Table 4: Comparison of different feature aggregation modules and scan strategies.

| Methods | Overall↓ | Acc.↓ | Comp.↓ | MAE↓ | GPU(G)↓ | Time(s)↓ | Params(M)↓ |
|---------|----------|-------|--------|------|---------|----------|------------|
| w/o Aggregation | 0.305 | 0.315 | 0.295 | 6.12 | 2.78 | 0.09 | 0.98 |
| w/ DCN [15] | 0.295 | 0.310 | 0.280 | 5.84 | 4.35 | 0.55 | 1.65 |
| w/ FMT [18] | 0.296 | 0.311 | 0.281 | 5.93 | 2.85 | 0.19 | 1.27 |
| w/ ET [24] | 0.291 | 0.310 | 0.272 | 5.62 | 2.91 | 0.17 | 1.09 |
| w/ VMamba [49] | 0.291 | 0.310 | 0.272 | 5.30 | 2.82 | 0.13 | 1.31 |
| w/ EVMamba [50] | 0.298 | 0.320 | 0.276 | 5.81 | 2.82 | 0.11 | 1.31 |
| w/ JamMa[54] | 0.301 | 0.318 | 0.284 | 6.01 | 2.82 | 0.11 | 1.31 |
| MVSMamba | 0.287 | 0.314 | 0.260 | 5.21 | 2.82 | 0.11 | 1.31 |

## 4.4 Ablation Study

We conducted ablation study (more in Appendix C) to analyze the effectiveness and efficiency of the proposed module using the metrics reported in Tab. 1, along with an additional depth metric, Mean Absolute Error (MAE). Unless otherwise specified, we use the model trained on DTU [58] low-resolution, evaluated with 5-view images at a resolution of $832 \times 1152$, with all other hyperparameters kept consistent. Since the point cloud metrics on the DTU dataset are highly sensitive to the depth fusion method and its hyperparameters, we provide additional quantitative ablation studies on detailed depth metrics in Appendix C.2 to further validate the effectiveness of our method.

**Effectiveness of Each Component.** As shown in Tab. 3, we ablate each component of our proposed method. The DM module contributes the most, as it capture both intra- and inter-view long-range dependencies between reference and source features at the bottom level of the FPN, and decodes them into all subsequent scales. The SDM modules perform self-feature enhancement at higher levels, strengthening multi-scale interactions and further improving performance. The MLP enhances the feature representations produced by Mamba blocks with only a slight increase in parameter count. All three modules incur minimal computational overhead, making our method highly efficient.

**Different Feature Aggregation Modules and Scan Strategies.** As shown in Tab. 4 row 2-4, we compare our method with three feature aggregation module: DCN [15, 13, 18], FMT [18], and ET [24]. Our method achieves the best performance improvement with minimal cost in memory and runtime, and only a modest growth in parameter count. As shown in Tab. 4 row 5-7, we compare our reference-centered dynamic scan strategy with three scan strategy: the four-directional scan used in VMamba[49], the skip scan used in EVMamba [50], and the joint scan used in JamMa [54]. Our proposed scan strategy achieves the best performance while maintaining the highest efficiency.

**Multi-Scale Aggregation.** We conducted an ablation study on multi-scale aggregation to evaluate the impact of applying the DM-module and SDM-module at different scales. As shown in Tab. 5, simply increasing the number of application scales for the DM-module or SDM-module does not yield further performance gains. This is because the DM-module, operating at the coarsest 0-th scale, already captures effective intra- and inter-view interactions. These interactions are then propagated through the decoder to all scales. Meanwhile, the SDM-module serves as a complement to the DM-module, providing self-feature enhancement. Therefore, given that the DM-module is applied at the 0-th scale, the SDM-module is applied at the 1-st scale instead.

Table 5: Ablation study on multi-scale aggregation in DM and SDM modules across FPN scales.

| Modules | Overall↓ | Acc.↓ | Comp.↓ | MAE↓ | GPU(G)↓ | Time(s)↓ | Params(M)↓ |
|---|---|---|---|---|---|---|---|
| DM (s=0) | 0.289 | 0.317 | 0.261 | 5.34 | 2.82 | 0.10 | 1.15 |
| DM (s=0,1) | 0.295 | 0.320 | 0.270 | 5.41 | 2.82 | 0.11 | 1.20 |
| DM (s=0,1,2) | 0.292 | 0.320 | 0.294 | 5.38 | 2.82 | 0.17 | 1.21 |
| DM (s=0,1,2,3) | 0.294 | 0.318 | 0.270 | 5.49 | 2.82 | 0.31 | 1.22 |
| DM (s=0) + SDM (s=1) | 0.287 | 0.314 | 0.260 | 5.21 | 2.82 | 0.11 | 1.31 |
| DM (s=0) + SDM (s=1,2) | 0.293 | 0.316 | 0.270 | 5.27 | 2.82 | 0.16 | 1.31 |
| DM (s=0) + SDM (s=1,2,3) | 0.296 | 0.318 | 0.274 | 5.35 | 3.38 | 0.35 | 1.31 |

Table 6: Ablation study of different feature concatenation.

| Methods | Overall(mm)↓ | Acc.(mm)↓ | Comp.(mm)↓ | MAE(mm)↓ |
|---|---|---|---|---|
| Source-centered static | 0.300 | 0.318 | 0.282 | 5.42 |
| Reference-centered static | 0.294 | 0.310 | 0.278 | 5.28 |
| Source-centered dynamic | 0.296 | 0.312 | 0.280 | 5.39 |
| Reference-centered dynamic | 0.287 | 0.314 | 0.260 | 5.23 |

Table 7: Ablation study on weight sharing.

| Sharing | Overall(mm)↓ | Acc.(mm)↓ | Comp.(mm)↓ | MAE(mm)↓ | Params(M)↓ |
|---|---|---|---|---|---|
| ✓ | 0.289 | 0.314 | 0.264 | 5.36 | 1.21 |
| ✗ | 0.287 | 0.314 | 0.260 | 5.21 | 1.31 |

**Different Feature Concatenation Methods.**   As shown in Table 6, we conducted an ablation study on four feature concatenation scanning methods: source-centered static, reference-centered static, source-centered dynamic, and reference-centered dynamic. The results show that our proposed reference-centered dynamic method achieves the best performance across both point cloud and depth metrics. We attribute this superior performance to the source features' ability to learn consistent global representations from the reference feature (Appendix D.1).

**Weight Sharing.**   As shown in Tab. 7, we conducted an ablation study to determine whether the four Mamba modules should share weights. Using four independent Mamba modules (with out weight sharing) achieves better performance. Due to Mamba's efficiency, this configuration with only a 0.1M increase in the parameter count. This indicates that independent Mamba modules allow different scanning directions to learn distinct information from the sequence, thereby improving model performance.

## 5   Conclusion

In this paper, we present a Mamba-based MVS network, termed MVSMamba, which efficiently aggregates global and omnidirectional feature representations. Specifically, we propose a DM-module with a novel reference-centered dynamic scanning strategy. This strategy enables anisotropic scanning from the reference feature to the source feature, where the scanning direction is dynamically updated based on the index of each source view to achieve omnidirectional coverage. The DM-module is integrated into the FPN to facilitate multi-scale feature aggregation. Experimental results demonstrate that our method outperforms state-of-the-art methods on multiple datasets while maintaining superior efficiency.

**Acknowledgments.** This work was supported by the Beijing Natural Science Foundation (No. L257003), National Natural Science Foundation of China (No. 62402042 and 62227801) and Fundamental Research Funds for the Central Universities (No. FRF-TP-25-033).

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

# A  Mamba

## A.1  State Space Models

State Space Models (SSMs) are originally designed to model continuous linear time-invariant systems [64]. These models map an input signal $x(t)$ to an output $y(t)$ via a hidden state $\mathbf{h}(t)$ as:

$$\mathbf{h}'(t) = \mathbf{A}\mathbf{h}(t) + \mathbf{B}x(t), \quad y(t) = \mathbf{C}\mathbf{h}'(t), \tag{7}$$

where $\mathbf{A} \in \mathbb{R}^{N \times N}$, $\mathbf{B} \in \mathbb{R}^{N \times 1}$, and $\mathbf{C} \in \mathbb{R}^{1 \times N}$ are system parameters. To enable the application of SSMs in discrete sequence modeling tasks, such as sequence-to-sequence learning, S4 [25] discretizes these parameters using the zero-order hold method. However, S4 shares parameters across all time steps, which limits its expressiveness in complex sequential contexts.

## A.2  Mamba Module

To address the limitations of S4, Mamba [26] introduces a refined formulation named S6, where the SSM parameters $\mathbf{B}$ and $\mathbf{C}$ are made input-dependent. This dynamic parameterization allows Mamba to adaptively modulate state transitions based on the input sequence, significantly enhancing its representation power and enabling performance on par with Transformer models [65]. Moreover, Mamba achieves high efficiency by reformulating the recurrent SSM computation into a single global convolution operation. Specifically, a convolution kernel $\mathbf{K}$ is precomputed, allowing output computation as:

$$\mathbf{K} = (\mathbf{CB}, \mathbf{CAB}, \dots, \mathbf{CA}^{N-1}\mathbf{B}), \quad y = x * \mathbf{K}, \tag{8}$$

where $*$ denotes the convolution operator. This structure supports both dynamic modeling and fast parallel training.

# B  More Quantitative Results

## B.1  Evaluation on ETH3D

ETH3D [66] benchmark contains high-resolution images with significant viewpoint transformations. We adopt an automatic evaluation process by uploading the generated point clouds to the official website. This process measures the accuracy (Acc.) and completeness (Comp.) of the generated point clouds. The F-score is defined as the harmonic mean of Acc. and Comp. We evaluate MVSMamba on the ETH3D training benchmark using the model finetuned on BlendedMVS [62], with the number of input views set to 11 and the image resolution to 1600×2432. However, point cloud fusion involves complex post-processing steps, requiring careful, per-scene hyperparameter selection to improve metrics. For a fair comparison, we follow the approach of MVSFormer++ [20]. We adopt the default dynamic fusion strategy [32] and set the depth confidence filtering threshold to 0.5 for all subscenes, without any hyperparameter tuning. As shown Tab. 8, MVSMamba achieved competitive performance with MVSFormer++, while also realizing a 52.1% reduction in running time and a 28.5% reduction in GPU memory consumption, thanks to the DM-module's efficient multi-view global feature representation. In contrast, Transformers result in impractically high complexity when processing such high-resolution images.

Table 8: Quantitative results on the ETH3D benchmark.

| Methods | Acc.(%)↑ | Comp.(%)↑ | F-score(%)↑ | Time(s)↓ | Memory(G)↓ |
|---------|----------|-----------|-------------|----------|------------|
| MVSFormer++ [20] | 81.88 | **83.88** | **82.99** | 2.11 | 9.31 |
| MVSMamba (Ours) | **87.87** | 76.85 | 81.69 (-1.5%) | **1.01** (-52.1%) | **6.65** (-28.5%) |

## B.2  Comparison with Feed-Forward MVS on DTU

DUSt3R [67] series of feed-forward MVS methods (such as MASt3R [68] and VGGT [69]) are trained on diverse datasets containing millions of images and perform 3D reconstruction without

known Ground-Truth (GT) cameras. In contrast, MVSNet-based [9] methods (such as MVSMamba and MVSFormer++) are trained solely on the DTU and BlendedMVS datasets and require known GT cameras to construct cost volumes. Due to these fundamental differences, these two categories of methods are not directly comparable. Tab. 9 nonetheless presents a direct performance comparison on DTU. MVSMamba significantly outperforms feed-forward methods that operate without known GT cameras (DUSt3R, VGGT), as well as MASt3R, which triangulates matches using known GT cameras to derive depth maps.

Table 9: Quantitative comparison with feed-forward MVS methods on the DTU dataset.

| Methods | Known GT camera | Overall(mm)↓ | Acc.(mm)↓ | Comp.(mm)↓ |
|---|---|---|---|---|
| DUSt3R [67] | ✗ | 1.741 | 2.677 | 0.805 |
| VGGT [69] | ✗ | 0.382 | 0.389 | 0.374 |
| MASt3R [68] | ✓ | 0.374 | 0.403 | 0.344 |
| MVSMamba(Ours) | ✓ | **0.280** | **0.308** | **0.252** |

## C    Additional Ablation Study

### C.1    Loss Function

As shown in Tab. 10, Cross-Entropy (CE) loss significantly outperforms $L_1$ loss on all point cloud metrics, while the difference in depth metrics is minimal. This is because CE loss directly supervises the probability volume, yielding more reliable confidence maps that are crucial for the subsequent depth map fusion process.

Table 10: Ablation study on loss function.

| Loss | Overall(mm)↓ | Acc.(mm)↓ | Comp.(mm)↓ | MAE(mm)↓ |
|---|---|---|---|---|
| $L_1$ | 0.302 | 0.319 | 0.285 | 5.21 |
| CE | 0.287 | 0.314 | 0.260 | 5.21 |

### C.2    Ablation on Depth Metrics

Consistent with the settings in the main paper Sec. 4.4, we conducted detailed ablation studies on depth metrics to further validate our method's effectiveness. These metrics include Mean Absolute Error (MAE), Root Mean Square Error (RMSE), and depth precision (Prec.) at thresholds of 1mm, 2mm, and 4mm. All metrics were evaluated at a resolution of 832×1152 on DTU [58]. Tab. 11 shows the effectiveness of each component. Tab. 12 compares different feature aggregation modules and scanning strategies. Tab. 13 compares different feature concatenation methods. Tab. 14 evaluates the impact of weight sharing among the four Mamba modules. Tab. 15 compares the performance of different loss functions.

Table 11: Ablation study of each component in MVSMamba.

| Modules | MAE(mm)↓ | RMSE(mm)↓ | Prec. 1mm(%)↑ | Prec. 2mm(%)↑ | Prec. 4mm(%)↑ |
|---|---|---|---|---|---|
| full | 9.59 | 27.78 | 66.01 | 78.06 | 84.09 |
| w/o DM | 11.29 | 30.72 | 64.16 | 76.46 | 82.58 |
| w/o SDM | 10.63 | 30.41 | 66.75 | 78.54 | 84.34 |
| w/o MLP | 8.72 | 26.04 | 65.58 | 77.89 | 83.96 |

### C.3    More Input Views

The DM-module adopts a reference-centered scanning strategy, allowing the reference features to fully leverage multi-view information for learning global and omnidirectional feature representations.

Table 12: Comparison of different feature aggregation modules and scan strategies.

| Methods | MAE(mm)↓ | RMSE(mm)↓ | Prec. 1mm(%)↑ | Prec. 2mm(%)↑ | Prec. 4mm(%)↑ |
|---|---|---|---|---|---|
| w/o Aggregation | 10.31 | 30.06 | 63.28 | 76.29 | 82.79 |
| w/ DCN [15] | 11.73 | 32.83 | 65.01 | 77.11 | 82.88 |
| w/ FMT [18] | 17.83 | 45.98 | 64.01 | 75.50 | 80.89 |
| w/ ET [24] | 13.24 | 35.34 | 65.52 | 77.33 | 83.08 |
| w/ VMamba [49] | 11.58 | 30.83 | 65.63 | 77.57 | 83.39 |
| w/ EVMamba [50] | 12.59 | 33.78 | 64.91 | 77.16 | 83.06 |
| w/ JamMa[54] | 15.89 | 40.08 | 57.53 | 71.95 | 79.49 |
| MVSMamba | 9.59 | 27.78 | 66.01 | 78.06 | 84.09 |

Table 13: Ablation study of different feature concatenation.

| Methods | MAE(mm)↓ | RMSE(mm)↓ | Prec. 1mm(%)↑ | Prec. 2mm(%)↑ | Prec. 4mm(%)↑ |
|---|---|---|---|---|---|
| Source-centered static | 12.11 | 33.09 | 64.07 | 76.90 | 82.97 |
| Reference-centered static | 9.97 | 27.97 | 63.93 | 76.90 | 83.15 |
| Source-centered dynamic | 10.21 | 28.67 | 64.54 | 76.99 | 83.22 |
| Reference-centered dynamic | 9.59 | 27.78 | 66.01 | 78.06 | 84.09 |

Table 14: Ablation study on weights sharing.

| Sharing | MAE(mm)↓ | RMSE(mm)↓ | Prec. 1mm(%)↑ | Prec. 2mm(%)↑ | Prec. 4mm(%)↑ |
|---|---|---|---|---|---|
| ✓ | 10.67 | 29.74 | 66.00 | 77.88 | 83.76 |
| ✗ | 9.59 | 27.78 | 66.01 | 78.06 | 84.09 |

Table 15: Ablation study on different loss function.

| Loss | MAE(mm)↓ | RMSE(mm)↓ | Prec. 1mm(%)↑ | Prec. 2mm(%)↑ | Prec. 4mm(%)↑ |
|---|---|---|---|---|---|
| $L_1$ | 9.27 | 25.64 | 67.60 | 78.01 | 83.60 |
| CE | 9.59 | 27.78 | 66.01 | 78.06 | 84.09 |

Table 16: Ablation study of the total number of training and testing views (reference and source views) on the Tanks-and-Temples [59] benchmark.

| Train | Test | Intermediate F-score [%] ↑ | | | | | | | | Advanced F-score [%] ↑ | | | | | | |
|---|---|---|---|---|---|---|---|---|---|---|---|---|---|---|---|---|
| | | Mean | Fam. | Fra. | Hor. | L.H. | M60 | Pan. | P.G. | Tra. | Mean | Aud. | Bal. | Cou. | Mus. | Pal. | Tem. |
| 7 | 21 | 65.00 | 81.07 | 70.85 | 49.83 | 68.00 | 63.81 | 64.84 | 64.58 | 57.67 | 39.28 | 24.60 | 44.33 | 36.96 | 51.82 | 36.50 | 41.99 |
| 9 | 21 | 66.46 | 82.01 | 72.30 | 52.89 | 69.49 | 64.29 | 65.98 | 65.58 | 59.08 | 42.27 | 28.84 | 48.98 | 39.73 | 53.87 | **36.80** | 45.42 |
| 11 | 21 | **67.67** | **82.47** | **72.90** | **58.55** | **69.63** | **65.34** | **66.88** | **65.60** | **59.98** | **43.32** | **30.95** | **49.61** | **41.04** | **54.92** | 36.72 | **46.67** |
| 11 | 11 | 65.90 | 82.43 | 70.55 | 55.63 | 66.33 | 65.00 | 64.59 | 63.83 | 59.02 | 41.82 | 29.81 | 46.96 | 39.61 | 52.65 | 36.48 | 45.39 |

To assess how our method benefits from the number of input views processed by the DM-module, we conduct an ablation study by varying the number of input views during both training and testing. As shown in Tab. 16, the performance consistently improves with more input views in both training and testing stages. The 20 candidate source views are extended by MVSFormer [30].

# D More Visualization Results

## D.1 PCA Features

We apply Principal Component Analysis (PCA) to reduce the number of feature channels to three and visualize the results with RGB. As illustrated in Fig. 6, we present the evolution of each reference-

source feature pair in the Courtroom scene from the Tanks-and-Temples Advanced set. The results show that all source features effectively learn consistent global representations from the reference feature, thereby facilitating more reliable subsequent feature matching.

## D.2   All Point Clouds

As shown Fig. 7 and Fig. 8, we visualize the reconstructed point clouds on the DTU [58] and Tanks-and-Temples [59] benchmark, respectively.

# E   Limitations

The proposed DM-module and SDM-module are effective when applied at specific FPN scales, simply extending them to FPN encoder features across multiple scales does not yield additional performance gains, indicating that the full potential of Mamba is not yet fully leveraged. Although the FPN structure allows global features to propagate from coarse to fine scales, this process inevitably introduces information loss. Developing a feature interaction framework that supports efficient multi-scale Mamba-based interaction remains a promising direction for future work.

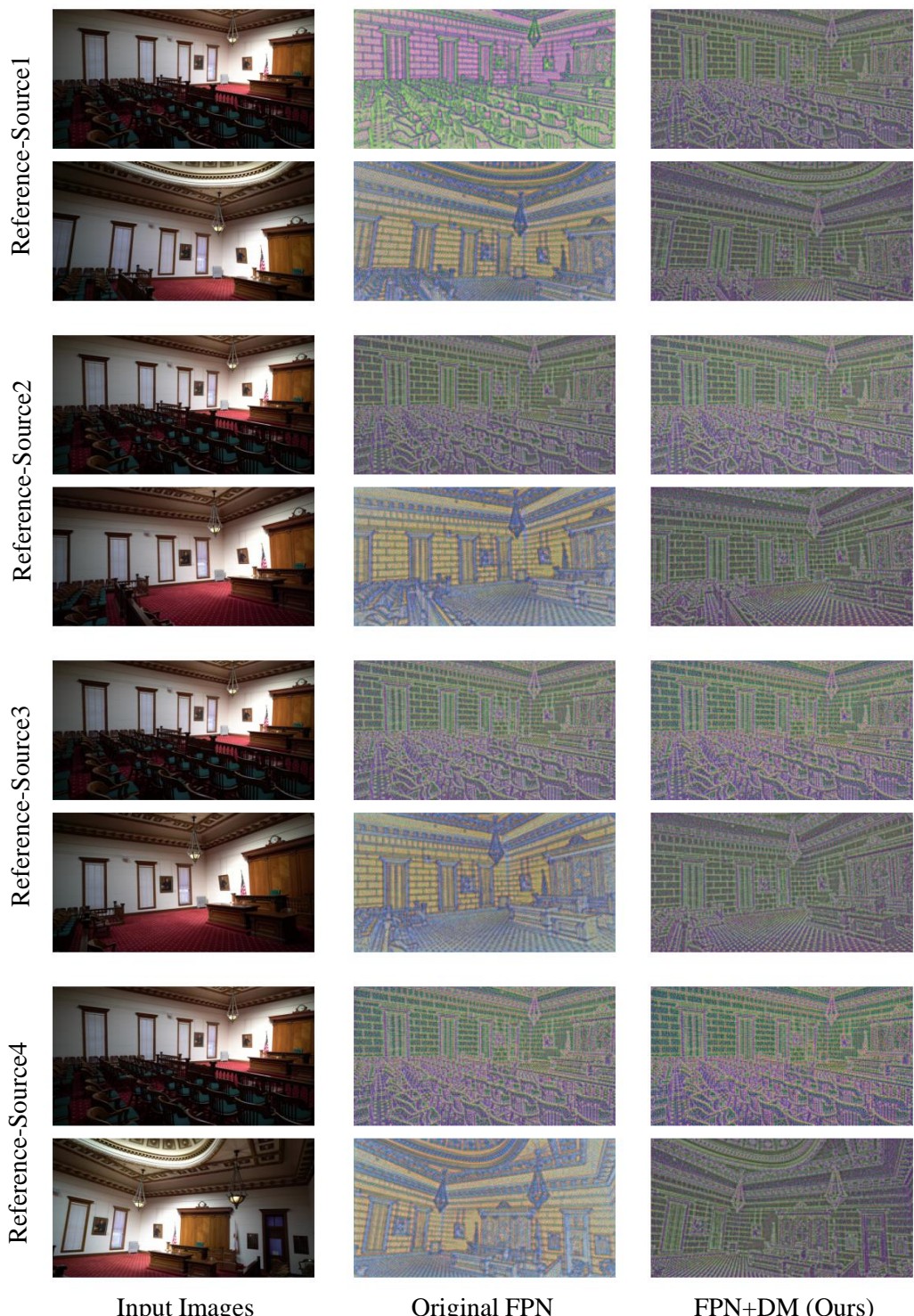

| Input Images | Original FPN | FPN+DM (Ours) |

Figure 6: We show the PCA features of each pair of reference-source features on the Courtroom scene of the Tanks-and-Temples [59] benchmark at the 0-th scale. For each pair, the top row displays the reference feature, while the bottom row shows the corresponding source feature. The source features are able to learn consistent global representations from the reference feature.

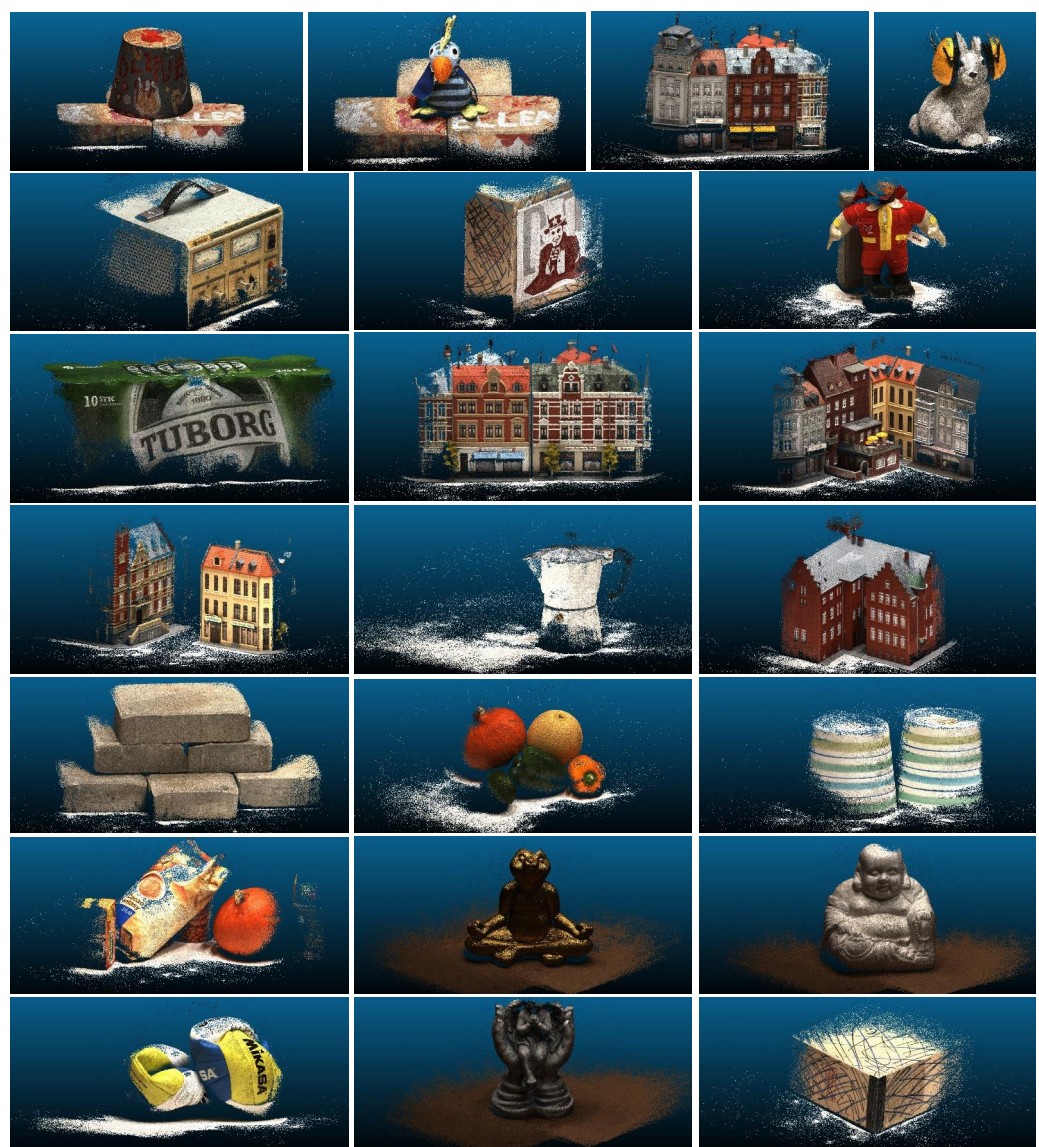

Figure 7: All reconstructed point clouds on the DTU [58] dataset by the proposed method.

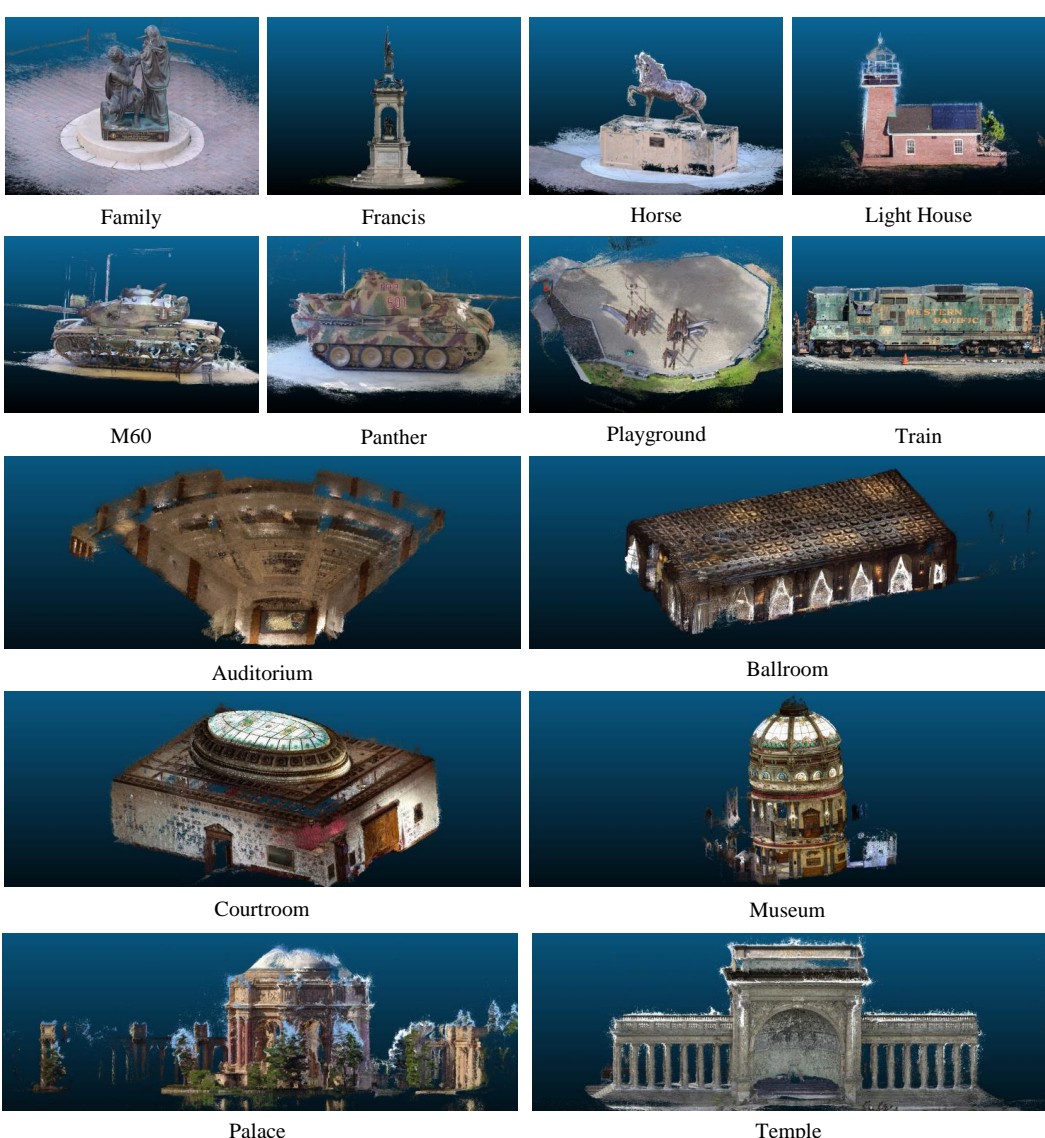

Figure 8: All reconstructed point clouds on the Tanks-and-Temples [59] benchmark by the proposed method.

