# OpenReview forum: "MVSMamba: Multi-View Stereo with State Space Model"
_NeurIPS.cc/2025/Conference — NeurIPS 2025 poster_

### Official Review · Reviewer_qVE2 · 2025-06-03

**Clarity:** 3
**Significance:** 2
**Originality:** 3
**Rating:** 4
**Confidence:** 4

**Summary:**

This paper proposes MVSMamba , a Mamba-based network for MVS. By leveraging the global modeling capability and linear complexity of the Mamba architecture, MVSMamba introduces a Dynamic Mamba module (DM-module) with a reference-centered dynamic scanning strategy to enable efficient feature interaction. Experimental results show that MVSMamba outperforms state-of-the-art methods on the DTU dataset and Tanks-and-Temples benchmark.

**Questions:**

1) The baseline without feature aggregation already performs well, and using VMamba instead of your DM-module gives a slightly better score (0.291 vs 0.287). Since the DTU results can be affected by many factors like post-processing, could you provide more detailed depth evaluation metrics (like RMSE or accuracy) to show that your method truly improves performance?

2) The authors concatenate source features around the reference view in four directions. Do these directions consider the actual positions of the views? Have you tested different ways of concatenation to see which one works best?

3) The authors use four independent Mamba blocks for the four directions. Why not share weights between them? Is there any analysis on whether they learn different or similar information?

**Ethical Concerns:**

["NO or VERY MINOR ethics concerns only"]

**Final Justification:**

The authors have addressed my concerns, and I find the paper’s contributions:  being among the first to apply Mamba to MVS, effectively modeling long-range dependencies with linear complexity, and achieving competitive performance with low computational cost, sufficient to meet the acceptance bar.

**Limitations:**

yes

**Quality:**

3

**Strengths And Weaknesses:**

Strengths:
1) This work introduces MVSMamba , which is among the first attempts to apply the Mamba architecture to the Multi-View Stereo (MVS) task. The method explores the potential of Mamba in modeling long-range dependencies with linear complexity, offering a new direction for global feature aggregation in MVS.
2) The experiments on the DTU dataset and Tanks-and-Temples benchmark show that MVSMamba achieves competitive performance compared to existing methods, while maintaining relatively low computational cost.

Weaknesses
1)  As shown in Table 4, the "w/o aggregation" baseline already achieves strong performance. Moreover, when alternative feature aggregation methods such as VMamba are applied, the model achieves an overall score of 0.291, which is even slightly better than the proposed MVSMamba's score of 0.287. Given that the overall score on the DTU dataset can be highly sensitive to various factors (including post-processing hyperparameters) it is unclear whether the proposed method truly brings a meaningful improvement. The authors are encouraged to provide more quantitative depth estimation metrics to better justify the effectiveness of MVSMamba.
2) The module employs directional feature concatenation (e.g., horizontal-right, horizontal-left, vertical-bottom, vertical-top) to facilitate scanning from the reference view to source views. However, it remains unclear whether this design considers the relative spatial positions between views. A quantitative analysis comparing different concatenation strategies and their impact on performance would strengthen the technical justification.
3) In the current design, four independent Mamba blocks are used to process the four directional sequences. The lack of weight sharing or cross-branch interaction mechanisms raises concerns about potential redundancy and limits the modeling of complementary information across directions.

---

> ### Author Rebuttal · Authors · 2025-07-31
>
> We thank the reviewer for the constructive comments. We provide our feedback as follows.
>
> > **Q1: As shown in Table 4, the "w/o aggregation" baseline already achieves strong performance. Moreover, when alternative feature aggregation methods such as VMamba are applied, the model achieves an overall score of 0.291, which is even slightly better than the proposed MVSMamba's score of 0.287. Given that the overall score on the DTU dataset can be highly sensitive to various factors (including post-processing hyperparameters) it is unclear whether the proposed method truly brings a meaningful improvement. The authors are encouraged to provide more quantitative depth estimation metrics to better justify the effectiveness of MVSMamba. The baseline without feature aggregation already performs well, and using VMamba instead of your DM-module gives a slightly better score (0.291 vs 0.287). Since the DTU results can be affected by many factors like post-processing, could you provide more detailed depth evaluation metrics (like RMSE or accuracy) to show that your method truly improves performance?**
>
> **A1:** It's important to clarify that the post-processing stage uses a hyperparameter-consistent dynamic fusion strategy with a confidence threshold of 0.55 for all scans in DTU. We provide detailed depth evaluation metrics (resolution: 832×1152) corresponding to the ablation study in Table 3 and Table 4 of the main paper (MAE is evaluated with a resolution of 512×640 during the validation stage) to show our methods truly improve depth performance.
>
> - The first table shows the effectiveness of each component of MVSMamba in terms of depth metrics, achieving the best MAE and RMSE without adding MLP, and the best depth precision without adding SDM.
> - The second table shows the comparison results of depth metrics with other feature aggregation methods and scanning methods. Our proposed MVSMamba achieved the best performance in all depth metrics.
>
> | Modules | MAE (mm)↓ | RMSE (mm)↓ | Precision 1mm (%)↑ | Precision 2mm (%)↑ | Precision 4mm (%)↑ |
> | :------ | :-------: | :--------: | :----------------: | :----------------: | :----------------: |
> | full    |   9.59    |   27.78    |       66.01        |       78.06        |       84.09        |
> | w/o DM  |   11.29   |   30.72    |       64.16        |       76.46        |       82.58        |
> | w/o SDM |   10.63   |   30.41    |     **66.75**      |     **78.54**      |     **84.34**      |
> | w/o MLP | **8.72**  | **26.04**  |       65.58        |       77.89        |       83.96        |
>
> | Methods         | MAE (mm)↓ | RMSE (mm)↓ | Precision 1mm (%)↑ | Precision 2mm (%)↑ | Precision 4mm (%)↑ |
> | --------------- | :-------: | :--------: | :----------------: | :----------------: | :----------------: |
> | w/o Aggregation |   10.31   |   30.06    |       63.28        |       76.29        |       82.79        |
> | w/ DCN          |   11.73   |   32.83    |       65.01        |       77.11        |       82.88        |
> | w/ FMT          |   17.83   |   45.98    |       64.01        |       75.50        |       80.89        |
> | w/ ET           |   13.24   |   35.34    |       65.52        |       77.33        |       83.08        |
> | w/ VMamba       |   11.58   |   30.83    |       65.63        |       77.57        |       83.39        |
> | w/ EVMamba      |   12.59   |   33.78    |       64.91        |       77.16        |       83.06        |
> | w/ JamMa        |   15.89   |   40.08    |       57.53        |       71.95        |       79.49        |
> | MVSMamba        | **9.59**  | **27.78**  |     **66.01**      |     **78.06**      |     **84.09**      |
>
> > **Q2: The module employs directional feature concatenation (e.g., horizontal-right, horizontal-left, vertical-bottom, vertical-top) to facilitate scanning from the reference view to source views. However, it remains unclear whether this design considers the relative spatial positions between views. A quantitative analysis comparing different concatenation strategies and their impact on performance would strengthen the technical justification. The authors concatenate source features around the reference view in four directions. Do these directions consider the actual positions of the views? Have you tested different ways of concatenation to see which one works best?**
>
> **A2:**
>
> - Our current method does not explicitly consider the actual positions of different camera views, which represents a potential direction for future improvement.
> - As shown in the table below, this study evaluates four feature concatenation scanning methods: source-centered static scanning, reference-centered static scanning, source-centered dynamic scanning, and our proposed reference-centered dynamic scanning. The statistical results show that our proposed reference-centered dynamic scanning method achieves the best performance in both point cloud metrics and depth metrics. This is because the source features learn consistent global feature representations from the reference features, enabling reliable feature matching, as shown in Figure 1 in the supplementary materials.
>
> | Scan Methods               | Overall (mm)↓ | Accuracy (mm)↓ | Completeness (mm)↓ | MAE (mm)↓ |
> | -------------------------- | :-----------: | :------------: | :----------------: | :-------: |
> | source-centered static     |     0.300     |     0.318      |       0.282        |   5.42    |
> | reference-centered static  |     0.294     |   **0.310**    |       0.278        |   5.28    |
> | source-centered dynamic    |     0.296     |     0.312      |       0.280        |   5.39    |
> | reference-centered dynamic |   **0.287**   |     0.314      |     **0.260**      | **5.21**  |
>
> | Scan Methods               | MAE (mm)↓ | RMSE (mm)↓ | Precision 1mm (%)↑ | Precision 2mm (%)↑ | Precision 4mm (%)↑ |
> | -------------------------- | :-------: | :--------: | :----------------: | :----------------: | :----------------: |
> | source-centered static     |   12.11   |   33.09    |       64.07        |       76.90        |       82.97        |
> | reference-centered static  |   9.97    |   27.97    |       63.93        |       76.90        |       83.15        |
> | source-centered dynamic    |   10.21   |   28.67    |       64.54        |       76.99        |       83.22        |
> | reference-centered dynamic | **9.59**  | **27.78**  |     **66.01**      |     **78.06**      |     **84.09**      |
>
> > **Q3: In the current design, four independent Mamba blocks are used to process the four directional sequences. The lack of weight sharing or cross-branch interaction mechanisms raises concerns about potential redundancy and limits the modeling of complementary information across directions. The authors use four independent Mamba blocks for the four directions. Why not share weights between them? Is there any analysis on whether they learn different or similar information?**
>
> **A3:** As the table below shows, ablation experiments determine whether the four Mamba modules share weights. The four independent Mamba modules that do not share weights achieve better performance. Due to Mamba's efficiency, this configuration increases the number of parameters by only 0.1M. This indicates that the four independent Mamba modules help different scanning directions learn different information from the sequence, thereby improving the model's performance.
>
> | Share Weights | Overall (mm)↓ | Accuracy (mm)↓ | Completeness (mm)↓ | MAE (mm)↓ | Params(M)↓ |
> | :------------ | :-----------: | :------------: | :----------------: | :-------: | :--------: |
> | √             |     0.289     |     0.314      |       0.264        |   5.36    |  **1.21**  |
> | ×             |   **0.287**   |   **0.314**    |     **0.260**      | **5.21**  |    1.31    |
>
> | Share Weights | MAE (mm)↓ | RMSE (mm)↓ | Precision 1mm (%)↑ | Precision 2mm (%)↑ | Precision 4mm (%)↑ |
> | ------------- | :-------: | :--------: | :----------------: | :----------------: | :----------------: |
> | √             |   10.67   |   29.74    |       66.00        |       77.88        |       83.76        |
> | ×             | **9.59**  | **27.78**  |     **66.01**      |     **78.06**      |     **84.09**      |

---

> > ### Comment · Reviewer_qVE2 · 2025-08-02
> >
> > Thank you for your response. I have reviewed it and am satisfied that my concerns have been adequately addressed. I appreciate the additional clarifications and hope the rebuttal discussions are reflected in the final version.

---

> > > ### Author Response · Authors · 2025-08-02
> > >
> > > Thank you for your valuable feedback. We're glad our response has addressed your concerns. The rebuttal discussions will be updated in the final version.

---

### Official Review · Reviewer_EQ7a · 2025-07-02

**Clarity:** 3
**Significance:** 2
**Originality:** 2
**Rating:** 4
**Confidence:** 3

**Summary:**

In MVS, good feature matching is key, but Transformer-based methods struggle with efficiency due to their quadratic complexity. To solve this, the authors introduce MVSMamba, which uses the Mamba architecture for efficient global feature aggregation with linear complexity.

The authors propose a Dynamic Mamba (DM) module, which scans features in multiple directions from the reference view. This enables efficient feature interaction across views, omnidirectional multi-view representations, and multi-scale feature aggregation, all while keeping the computational cost low.

**Questions:**

Given that ETH3D is a standard benchmark in the MVS community, could you clarify why this dataset was not included? Including ETH3D would provide a more comprehensive evaluation of MVSMamba’s generalizability across different datasets.

The paper does not compare MVSMamba with other recent methods such as DUST3R, VGGT, CUT3R, and FAST3R, which are important works in the MVS field. Could you include a comparison with these methods to better contextualize the performance of MVSMamba?

The paper appears to combine Mamba with MVSformer++.  I look forward to hearing other reviewers' perspectives and the authors' response regarding the core novelty.

**Ethical Concerns:**

["NO or VERY MINOR ethics concerns only"]

**Final Justification:**

The rebuttal has addressed my concerns. I have also read the reviews and rebuttals from the other reviewers. It appears that others recognize the contribution of introducing Mamba into MVS, as well as the design of the dynamic Mamba module, as valuable contributions. Based on this, I will raise my rating. I encourage the authors to include results on the ETH3D dataset, along with comparisons to MASt3R, DUSt3R, and VGGT, in the camera-ready version. I also look forward to seeing ETH3D results compared with MASt3R, DUSt3R, and VGGT.

**Limitations:**

yes

**Quality:**

3

**Strengths And Weaknesses:**

## Strengths

### Performance:

The experiments show that MVSMamba outperforms state-of-the-art MVS methods on both the DTU dataset and the Tanks-and-Temples benchmark, excelling in both accuracy and efficiency. The comparison with CNN-based and Transformer-based methods is particularly convincing, with MVSMamba showing the best performance and efficiency across the board.

### Clarity:

Figure 1 effectively illustrates the performance vs. efficiency trade-offs, clearly highlighting the superiority of MVSMamba over existing approaches. The visual comparison between CNN-based, Transformer-based, and Mamba-based methods is straightforward and well-executed.

## Weaknesses

### Limited Dataset Evaluation:

The paper lacks evaluation on the widely used ETH3D dataset.

### Lack of Comparison with Key Methods:

There is no comparison with important works such as DUST3R, VGGT, CUT3R, and FAST3R, which would help contextualize MVSMamba’s performance within the current landscape.

### Novelty:

While the approach is interesting, the combination of Mamba and MVSformer++ feels incremental.

---

> ### Author Rebuttal · Authors · 2025-07-31
>
> We thank the reviewer for the constructive comments. We provide our feedback as follows.
>
> > **Q1: The paper lacks evaluation on the widely used ETH3D dataset. Given that ETH3D is a standard benchmark in the MVS community, could you clarify why this dataset was not included? Including ETH3D would provide a more comprehensive evaluation of MVSMamba’s generalizability across different datasets.**
>
> **A1:** We use the model finetuned on BlendedMVS to evaluate MVSMamba on the large-scale scene dataset ETH3D, which consists of high-resolution images. We set the number of input views to 11 and the image resolution to 1600×2432. However, point cloud fusion involves complex post-processing steps, requiring careful selection of hyperparameters on a per-scene basis to improve metrics. For a fair comparison, we follow the approach of MVSFormer++ [1], adopting the default dynamic fusion strategy and setting the depth confidence filtering threshold to 0.5 for all subscenes, without any hyperparameter tuning. The evaluation results are shown in the table below.
>
> | Methods         | Accuracy (%)↑ | Completeness (%)↑ | F-score (%)↑  |     Time(s)↓      |    Memory(G)↓     |
> | :-------------- | :-----------: | :---------------: | :-----------: | :---------------: | :---------------: |
> | MVSFormer++ [1] |     81.88     |     **83.88**     |   **82.99**   |       2.11        |       9.31        |
> | MVSMamba (Ours) |   **87.87**   |       76.85       | 81.69 (-1.5%) | **1.01** (-52.1%) | **6.65** (-28.5%) |
>
> Our proposed MVSMamba achieved competitive performance to MVSFormer++, with a 52.1% reduction in running time and a 28.5% reduction in GPU memory consumption, thanks to the DM-module's efficient multi-view global feature representation. Transformers result in impractically high complexity when processing high-resolution images.
>
> > **Q2: There is no comparison with important works such as DUST3R, VGGT, CUT3R, and FAST3R, which would help contextualize MVSMamba’s performance within the current landscape. The paper does not compare MVSMamba with other recent methods such as DUST3R, VGGT, CUT3R, and FAST3R, which are important works in the MVS field. Could you include a comparison with these methods to better contextualize the performance of MVSMamba?**
>
> **A2:** DUSt3R [2] series methods are trained on a mixture of various datasets containing millions of images and perform 3D reconstruction without known GT cameras. In contrast, the MVSNet [3] series of methods (such as MVSMamba, MVSFormer++) are trained only on the DTU and BlendedMVS datasets and require known GT cameras to construct cost volumes. Generally speaking, the above two methods cannot be compared fairly.
>
> Below is a direct comparison of performance on DTU. MVSMamba's performance is significantly better than unknown GT camera methods (DUSt3R, VGGT [4]), as well as MASt3R [5], which derives depth maps by triangulating matches using known GT cameras. CUT3R [6] and Fast3R [7] do not report reconstruction metrics on DTU.
>
> | Methods         | Known GT camera | Overall (mm)↓ | Accuracy (mm)↓ | Completeness (mm)↓ |
> | --------------- | :-------------: | :-----------: | :------------: | :----------------: |
> | MVSMamba (Ours) |        √        |   **0.280**   |   **0.308**    |     **0.252**      |
> | MASt3R [5]      |        √        |     0.374     |     0.403      |       0.344        |
> | DUSt3R [2]      |        ×        |     1.741     |     2.677      |       0.805        |
> | VGGT [4]        |        ×        |     0.382     |     0.389      |       0.374        |
>
> > **Q3: While the approach is interesting, the combination of Mamba and MVSformer++ feels incremental. The paper appears to combine Mamba with MVSformer++. I look forward to hearing other reviewers' perspectives and the authors' response regarding the core novelty.**
>
> **A3:** MVSMamba is not a combination of Mamba and MVSFormer++. Both MVSMamba and MVSFormer++ are built on a pure four-stage coarse-to-fine framework similar to CasMVSNet [8]. Building on this, MVSFormer++ introduces several Transformer-based modules to enhance performance, such as incorporating pre-trained ViT to enhance CNN features, but with lower efficiency. In contrast, MVSMamba is the first model to introduce Mamba into the MVS domain, achieving outstanding performance and efficiency through the design of the dynamic Mamba module (DM-module).
>
> [1] Cao, Chenjie, et al. "Mvsformer++: Revealing the devil in transformer's details for multi-view stereo." ICLR 2024.
>
> [2] Wang, Shuzhe, et al. "Dust3r: Geometric 3d vision made easy." CVPR 2024.
>
> [3] Yao, Yao, et al. "Mvsnet: Depth inference for unstructured multi-view stereo." ECCV 2018.
>
> [4] Wang, Jianyuan, et al. "Vggt: Visual geometry grounded transformer." CVPR 2025.
>
> [5] Leroy, Vincent, et al. "Grounding image matching in 3d with mast3r." ECCV 2024.
>
> [6] Wang, Qianqian, et al. "Continuous 3d perception model with persistent state." CVPR 2025.
>
> [7] Yang, Jianing, et al. "Fast3r: Towards 3d reconstruction of 1000+ images in one forward pass." CVPR 2025.
>
> [8] Gu, Xiaodong, et al. "Cascade cost volume for high-resolution multi-view stereo and stereo matching." CVPR 2020.

---

> > ### Comment · Reviewer_EQ7a · 2025-08-06
> >
> > Thank you for the rebuttal — it has addressed my concerns. I have also read the reviews and rebuttals from the other reviewers. It appears that others recognize the contribution of introducing Mamba into MVS, as well as the design of the dynamic Mamba module, as valuable contributions. Based on this, I will raise my rating.
> >
> > I encourage the authors to include results on the ETH3D dataset, along with comparisons to MASt3R, DUSt3R, and VGGT, in the camera-ready version. I also look forward to seeing ETH3D results compared with MASt3R, DUSt3R, and VGGT.

---

> > > ### Author Response · Authors · 2025-08-06
> > >
> > > Thank you for your valuable feedback. We're glad our response has addressed your concerns. The rebuttal results will be include in the final version.

---

### Official Review · Reviewer_iftD · 2025-07-02

**Clarity:** 3
**Significance:** 3
**Originality:** 3
**Rating:** 4
**Confidence:** 4

**Summary:**

The paper proposes a Mamba-based MVS network, to alleviate the quadratic computational complexity of Transformer-based methods while achieving comparable or even outperforming the Transformer-based methods. Specifically, it introduces a Dynamic Mamba Module based on a reference-centered dynamic scanning strategy. The model achieves good results on DTU dataset and Tanks-and-Temples dataset.

**Questions:**

1. Have you tried any real-world captures (maybe just 1-2 examples) and how the results look like?
2. It would be helpful to have more analysis on how the proposed modules help in difficult scenarios, e.g. textureless regions, highly-reflected regions etc.
3. Is there any failure cases and any analysis?
4. Wondering if you tried using expectation instead of winner-take-all when predicting depth. Also many previous methods e.g. CasMVSNet use L1 loss directly. Have you tried?

I am happy to increase the score if the questions above are addressed properly.

**Ethical Concerns:**

["NO or VERY MINOR ethics concerns only"]

**Final Justification:**

I have read reviews from other reviewers. The rebuttal addressed my questions. So I'll keep my score.

**Limitations:**

Yes, in the supplementary material.

**Paper Formatting Concerns:**

No.

**Quality:**

3

**Strengths And Weaknesses:**

Strengths:
1. It's the first paper to use Mamba in feature extraction for Multi-view Stereo, to consider long-range dependencies with linear time complexity.
2.  The proposed method achieves good results on both DTU and Tanks-and-Temples when only trained on DTU and finetuned on BlendedMVS.

Weaknesses:
1. I am wondering if the method can be applied to real-world captures where the poses might have noises (e.g. from COLMAP), and how robust the method can be?
2. The paper claims the method is both accurate and efficient. While it's easy to understand it's high efficiency, there's no analysis / visualization on how Mamba can help with the accuracy, e.g. for textureless regions, or for failure cases of other methods.
3. No discussion of failure cases.

---

> ### Author Rebuttal · Authors · 2025-07-31
>
> We thank the reviewer for the constructive comments. We provide our feedback as follows.
>
> > **Q1: I am wondering if the method can be applied to real-world captures where the poses might have noises (e.g. from COLMAP), and how robust the method can be? Have you tried any real-world captures (maybe just 1-2 examples) and how the results look like?**
>
> **A1:** Due to the rebuttal format's limitations, new visual results cannot be submitted at this stage. However, a dedicated section with qualitative results on real-world captures is included in the final version to further demonstrate the method's applicability.
>
> >**Q2: The paper claims the method is both accurate and efficient. While it's easy to understand it's high efficiency, there's no analysis / visualization on how Mamba can help with the accuracy, e.g. for textureless regions, or for failure cases of other methods. It would be helpful to have more analysis on how the proposed modules help in difficult scenarios, e.g. textureless regions, highly-reflected regions etc.**
>
> **A2:** The proposed modules are particularly beneficial in challenging regions (e.g., textureless, reflective) because they provide global contextual information that local CNN features inherently lack. This global context is crucial for disambiguating matches in low-information areas.
> While previous methods used computationally expensive Transformers for this purpose, our DM-module leverages Mamba's linear-complexity scanning to achieve efficient global feature aggregation across all views. As qualitatively shown in Figure 1 of our supplementary material, this allows source features to learn globally consistent representations from the reference feature, leading to more reliable feature matching.
> To make this point more explicit, we will add new visualizations in the final version that directly compare our method with and without the DM-module for these difficult scenarios. This will clearly illustrate our contribution.
>
> > **Q3: No discussion of failure cases. Is there any failure cases and any analysis?**
>
> **A3:** MVSMamba can aggregate global features with extremely high efficiency, but it still cannot effectively handle areas with extreme viewpoint changes or strong occlusions. In these areas, the fundamental geometric correspondence between views is lost,  our global feature aggregation cannot compensate for a complete lack of corresponding visual information. We will add visualizations and related analysis of these failure cases in the final version.
>
> > **Q4: Wondering if you tried using expectation instead of winner-take-all when predicting depth. Also many previous methods e.g. CasMVSNet use L1 loss directly. Have you tried?**
>
> **A4:** We have indeed conducted experiments comparing L1 loss with our current approach using Cross-Entropy (CE) loss. The statistical results for L1 loss and cross-entropy (CE) loss are as follows.
>
> - The first table shows the point cloud metrics and depth metrics (512×640) during the validation process.
> - The second table shows the detailed metrics of the depth map at a resolution of 832×1152.
>
> As the results show, CE loss significantly outperforms L1 loss in all point cloud metrics, while the difference in depth metrics is minimal. This is because CE loss directly supervises the probability volume, enabling more reliable confidence maps, which are crucial for subsequent depth map fusion processes. Therefore, recent SOTA MVS methods adopt a winner-take-all approach for depth prediction and CE loss for supervising the probability volume (e.g., MVSFormer++ [1] and GoMVS [2]).
>
> | Loss | Overall (mm)↓ | Accuracy (mm)↓ | Completeness (mm)↓ | MAE (mm)↓ |
> | ---- | :-----------: | :------------: | :----------------: | :-------: |
> | L1   |     0.302     |     0.319      |       0.285        |   5.21    |
> | CE   |   **0.287**   |   **0.314**    |     **0.260**      | **5.21**  |
>
> | Loss | MAE (mm)↓ | RMSE (mm)↓ | Precision 1mm (%)↑ | Precision 2mm (%)↑ | Precision 4mm (%)↑ |
> | :--- | :-------: | :--------: | :----------------: | :----------------: | :----------------: |
> | L1   | **9.27**  | **25.64**  |     **67.60**      |       78.01        |       83.60        |
> | CE   |   9.59    |   27.78    |       66.01        |     **78.06**      |     **84.09**      |
>
> [1] Cao, Chenjie, et al. "Mvsformer++: Revealing the devil in transformer's details for multi-view stereo." ICLR 2024.
>
> [2] Wu, Jiang, et al. "Gomvs: Geometrically consistent cost aggregation for multi-view stereo." CVPR 2024.

---

> > ### Comment · Reviewer_iftD · 2025-08-08
> >
> > I want to thank the authors for their rebuttal. I also read reviews from other reviewers. The rebuttal addressed my questions.

---

> > > ### Author Response · Authors · 2025-08-08
> > >
> > > Thank you for your valuable feedback. We're glad our explanation has addressed your questions. Your insights are greatly appreciated and help us further improve our work.

---

### Official Review · Reviewer_A2fC · 2025-07-03

**Clarity:** 2
**Significance:** 2
**Originality:** 2
**Rating:** 4
**Confidence:** 3

**Summary:**

This paper proposes a learning-based MVS method that utilizes the Mamba architecture. As the original Mamba does, the proposed method replaces the Transformer to be more efficient and faster. Additionally, the method includes a scanning strategy that considers multi-view images. Based on the features extracted from multiple Mamba modules, the depth is estimated coarse-to-fine and is trained with cross-entropy loss. The evaluation is done in the DTU and T&T datasets, showing competitive result metrics.

**Questions:**

I expect the authors to answer the rebuttal for the weakness, along with the following considerations: I'm still not convinced of the meaning of omnidirectional in the paper. It seems the omnidirectional means multiple scanning directions for rearranging image patches for Mamba input, not related to the view directions or other directional components. If so, it causes confusion for some authors, and I recommend replacing it for clarity.

Minor typo:
- [17] MVSformer++ reference missing authors

**Ethical Concerns:**

["NO or VERY MINOR ethics concerns only"]

**Final Justification:**

The author's rebuttal clarified my concerns. And the answers to the questions that the other reviewers raised also seem meaningful. Thus, I keep my ratings.

**Limitations:**

yes

**Paper Formatting Concerns:**

No formatting issues.

**Quality:**

3

**Strengths And Weaknesses:**

The paper shows one way to utilize the latest architecture for MVS. Since this is an initial attempt at using Mamba for the MVS problem, it can be shown as the paper's contribution. The proposed Dynamic Mamba module, including the scanning method, seems novel. However, its key idea is adopted from the existing Mamba-based computer vision papers (VMamba, MambaVision, EfficientViM, Mamba-ND, ...), and the remaining contribution is an extension to the multi-view images. As a result, it seems the contributions are relatively weak.
Except for the contribution part, the paper provides sufficient details of experiments and information. The evaluation result seems to outperform/comparable to the latest works, and ablation studies seem reasonable.

---

> ### Author Rebuttal · Authors · 2025-07-31
>
> We thank the reviewer for the constructive comments. We provide our feedback as follows.
>
> > **Q1: The proposed Dynamic Mamba module, including the scanning method, seems novel. However, its key idea is adopted from the existing Mamba-based computer vision papers (VMamba, MambaVision, EfficientViM, Mamba-ND, ...), and the remaining contribution is an extension to the multi-view images. As a result, it seems the contributions are relatively weak.**
>
> **A1:** We agree that our work builds upon the foundational success of Mamba in computer vision, a field pioneered by methods like VMamba [1]. Indeed, VMamba is instrumental in adapting Mamba for computer vision by introducing four-directional scanning to bridge the gap between 1D sequences and 2D images. Subsequent works, such as MambaVision [2], EfficientViM [3], and Mamba-ND [4], further explore this space by combining Mamba with self-attention, reducing computational costs, or extending the architecture to multi-dimensional data. These methods are designed for individual features and do not establish connections between multi-view features.
>
> However, our contribution is not a straightforward extension but a novel adaptation specifically designed for the unique challenges of Multi-View Stereo (MVS). Unlike the general-purpose methods mentioned, our work is the first to integrate the scanning mechanism with the specific characteristics of MVS feature matching. The core of our novelty lies in the Dynamic Mamba module (DM-module) and its reference-centered dynamic scanning strategy. This strategy is tailored for the "one-to-many" feature matching paradigm inherent in MVS. By concatenating a source feature around the reference feature and dynamically adjusting the scan directions for different source views, our approach facilitates rich and efficient intra- and inter-view feature interaction from the reference to the source views. This problem-specific design enables highly efficient feature aggregation.
>
> We include a more detailed discussion of other computer vision Mamba papers in the final version.
>
> > **Q2: I'm still not convinced of the meaning of omnidirectional in the paper. It seems the omnidirectional means multiple scanning directions for rearranging image patches for Mamba input, not related to the view directions or other directional components. If so, it causes confusion for some authors, and I recommend replacing it for clarity.**
>
> **A2:** We understand the concern regarding the meaning of "omnidirectional" in the paper. The term refers to the multiple scanning directions used to rearrange image patches for Mamba's input, rather than being related to camera viewpoints or other directional components.
>
> The impression that our method involves "rearranging" image patches is correct in a sense. This perspective arises from imagining a predefined indexing of patches (e.g., numbering them from top-left to bottom-right) and then creating different sequences by shuffling these indices.
>
> However, we clarify a nuance in our approach. Rather than reordering a fixed set of indices, our method directly changes the scanning path across the 2D feature map itself. The original Mamba architecture is designed for unidirectional 1D sequences. To make it understand 2D images, it is necessary to convert 2D features into 1D sequences. Our "omnidirectional scanning" achieves this by performing four independent scans from the four cardinal directions (i.e., top-to-bottom, bottom-to-top, left-to-right, and right-to-left), which generates four distinct 1D sequences. Because each path represents a different fundamental direction of traversal, we describe the process as "omnidirectional." This allows the model to perceive and aggregate features from all directions, effectively bridging the gap between a 1D sequence model and 2D spatial information.
>
> This principle of adjusting the information processing order to suit a specific task is common in computer vision. For example, while a standard Vision Transformer [5] processes patches in a linear sequence, the Swin Transformer [6] introduces "shifted windows" to selectively reorganize the interaction scope of patches, which allows it to capture richer contextual information and achieve superior results. Analogously, our method is a novel mechanism, specifically designed for the MVS task, that uses this omnidirectional scanning to enable Mamba to efficiently aggregate global multi-view features.
>
> [1] Liu, Yue, et al. "Vmamba: Visual state space model." NeurIPS 2024.
>
> [2] Hatamizadeh, Ali, et al. "Mambavision: A hybrid mamba-transformer vision backbone." CVPR 2025.
>
> [3] Lee, Sanghyeok, et al. "Efficientvim: Efficient vision mamba with hidden state mixer based state space duality." CVPR 2025.
>
> [4] Li, Shufan, et al. "Mamba-nd: Selective state space modeling for multi-dimensional data." ECCV 2024.
>
> [5] Dosovitskiy, Alexey, et al. "An Image is Worth 16x16 Words: Transformers for Image Recognition at Scale." ICLR 2021.
>
> [6] Liu, Ze, et al. "Swin transformer: Hierarchical vision transformer using shifted windows." ICCV 2021.

---

> > ### Comment · Reviewer_A2fC · 2025-08-06
> >
> > Thanks to the authors for clarifying my concerns. Both answers for the dynamic Mamba module's contribution and the meaning of omnidirectional make sense. I expect authors will include them in the paper if accepted.

---

> > > ### Author Response · Authors · 2025-08-06
> > >
> > > Thank you for your valuable feedback. We're glad our response has addressed your concerns. The rebuttal discussions will be include in the final version.

---

### Decision · Program_Chairs · 2025-09-17

**Decision:**

Accept (poster)

**Comment:**

To fully exploit Mamba’s potential in MVS, the paper proposed a Dynamic Mamba module (DM-module) based on a reference-centered
 dynamic scanning strategy, which enables: (1) Efficient intra- and inter-view feature interaction from the reference to source views, (2) Omnidirectional multi-view feature representations, and (3) Multi-scale global feature aggregation.

The major weaknesses lie in 1) incremental contribution by adopting existing Mamba-based work to multi-view stereo; 2) real-world evaluation; 3) comparison with very recent structure-from-motion pipeline such as VGGT. The authors provided rebuttal to address these comments in details, which successfully convince the reviewers.

Given the consistent 4XBorderline accept ratings from the reviewers, I would like to suggest to accept the paper. Nevertheless, the authors are requested to include all the deatils during the rebuttal period to the revised version.